# Association between hepatic oxygenation on near-infrared spectroscopy and clinical factors in patients undergoing hemodialysis

Yuichiro Ueda[1⊙], Susumu Ookawara[1⊙]*, Kiyonori Ito[1⊙], Yusuke Sasabuchi[2], Hideyuki Hayasaka[3], Masaya Kofuji[3], Takayuki Uchida[3], Sojiro Imai[4], Satoshi Kiryu[4], Saori Minato[1], Haruhisa Miyazawa[1], Hidenori Sanayama[5], Keiji Hirai[1], Kaoru Tabei[4], Yoshiyuki Morishita[1]

1 Division of Nephrology, Department of Integrated Medicine, Saitama Medical Center, Jichi Medical University, Saitama, Japan, 2 Data Science Center, Jichi Medical University, Tochigi, Japan, 3 Department of Clinical Engineering, Saitama Medical Center, Jichi Medical University, Saitama, Japan, 4 Department of Dialysis, Minami-Uonuma City Hospital, Niigata, Japan, 5 Division of Neurology, Department of Integrated Medicine, Saitama Medical Center, Jichi Medical University, Saitama, Japan

⊙ These authors contributed equally to this work.
* su-ooka@hb.tp1.jp

**Data Availability Statement:** All relevant data are within the manuscript and its Supporting information files.

## Abstract

The hepato-splanchnic circulation directly influences oxygenation of the abdominal organs and plays an important role in compensating for the blood volume reduction that occurs in the central circulation during hemodialysis (HD) with ultrafiltration. However, the hepato-splanchnic circulation and oxygenation cannot be easily evaluated in the clinical setting of HD therapy. We included 185 HD patients and 15 healthy volunteers as the control group in this study. Before HD, hepatic regional oxygen saturation ($rSO_2$), a marker of hepatic oxygenation reflecting the hepato-splanchnic circulation and oxygenation, was monitored using an INVOS 5100c oxygen saturation monitor. Hepatic $rSO_2$ was significantly lower in patients undergoing HD than in healthy controls ($56.4 \pm 14.9\%$ vs. $76.2 \pm 9.6\%$, $p < 0.001$). Multivariable regression analysis showed that hepatic $rSO_2$ was independently associated with body mass index (BMI; standardized coefficient: 0.294), hemoglobin (Hb) level (standardized coefficient: 0.294), a history of cardiovascular disease (standardized coefficient: -0.157), mean blood pressure (BP; standardized coefficient: 0.154), and serum albumin concentration (standardized coefficient: 0.150) in Model 1 via a simple linear regression analysis. In Model 2 using the colloid osmotic pressure (COP) in place of serum albumin concentration, the COP (standardized coefficient: 0.134) was also identified as affecting hepatic $rSO_2$. Basal hepatic oxygenation before HD might be affected by BMI, Hb levels, a history of cardiovascular disease, mean BP, serum albumin concentration, and the COP. Further prospective studies are needed to clarify whether changes in these parameters, including during HD, affect the hepato-splanchnic circulation and oxygenation in HD patients.

**Funding:** This work was supported by a grant from the Japanese Association of Dialysis Physicians (JADP Grant 2018-10, www.touseki-ikai.or.jp/htm/03_research/) and JSPS KAKENHI (no. JP20K11534, https://www-shinsei.jsps.go.jp/kaken/english/index.html) to SO. The funders of this study had no role in the study design; data collection, analysis, and interpretation; writing; or decision to submit the manuscript for publication.

**Competing interests:** The authors have no conflicts of interest to declare.

## Introduction

Body fluid management is an important aspect of hemodialysis (HD), and ultrafiltration is essential in achieving each patient's target body weight. During HD with ultrafiltration, blood shifting from the hepato-splanchnic circulation to the central circulation plays an important role in compensating for the blood volume reduction and decrease in cardiac output and blood pressure (BP) [1,2], in addition to fluid moving from the interstitium to the intravascular space. Therefore, the hepato-splanchnic circulation in patients undergoing HD has been attracting increasing attention [3,4]. However, the hepato-splanchnic circulation status cannot be easily evaluated in the clinical setting of HD therapy.

Near-infrared spectroscopy has been used to measure regional oxygen saturation ($rSO_2$), a marker of tissue oxygenation [5,6], to detect imbalances between arterial oxygen delivery and tissue oxygen consumption. The hepatic circulation consists of two different blood supplies, one from the hepatic artery and the other from the portal vein [7] and monitoring hepatic oxygenation would help capture important changes in oxygen distribution to the hepato-splanchnic circulation [8]. Hepatic $rSO_2$ values, which were recently used to evaluate the hepato-splanchnic circulation and oxygenation of patients undergoing HD, were reportedly maintained during HD without intradialytic hypotension [9], and significantly increased in response to an increase in hemoglobin (Hb) level by intradialytic blood transfusion [10]. Additionally, a decrease in hepatic $rSO_2$ was confirmed prior to intradialytic hypotension during HD [11,12]. However, few reports have examined the association between hepatic $rSO_2$ before HD and clinical factors in patients undergoing HD, and the clinical factors that affect hepatic $rSO_2$ remain unknown. Clarification of these factors might provide guidance for maintaining or improving patient hepato-splanchnic circulation and oxygenation status in the clinical setting of HD therapy. Therefore, this study aimed to elucidate the clinical factors influencing hepatic $rSO_2$ in patients undergoing HD.

## Materials and methods

### Participants

This study was performed at two facilities, including our hospital. Patients who met the following criteria were enrolled: (i) age > 20 years; (ii) end-stage renal disease managed with HD; (iii) started HD at least one month before the study; (iv) tissue thickness ≤ 20 mm from the skin to the surface of the liver in the right intercostal area as measured by ultrasonography; and (v) hepatic $rSO_2$ data collected using an INVOS 5100c oxygen saturation monitor. The exclusion criteria were coexisting major diseases, including congestive heart failure or neurological disorders, such as severe cerebrovascular disease and cognitive impairment.

Fig 1 shows a flow chart of patient enrollment and analysis. Of the 277 patients screened, 224 met the inclusion criteria and were enrolled between August 1, 2013 and December 31, 2019. Overall, 39 patients were excluded from the analysis because of lack of data. Ultimately, 185 patients (24 from Minami-Uonuma City Hospital and 161 from our hospital) were included and analyzed in the present study. In addition, 15 healthy volunteers (nine men and six women; mean age, 38.2 ± 17.8 years) were recruited as a control group.

### Ethics approval

All participants provided written informed consent. The study was approved by the Institutional Review Board of the Saitama Medical Center at Jichi Medical University (Saitama, Japan: approval numbers, RIN 15–104 and RINS19-HEN007) and Minami-Uonuma City

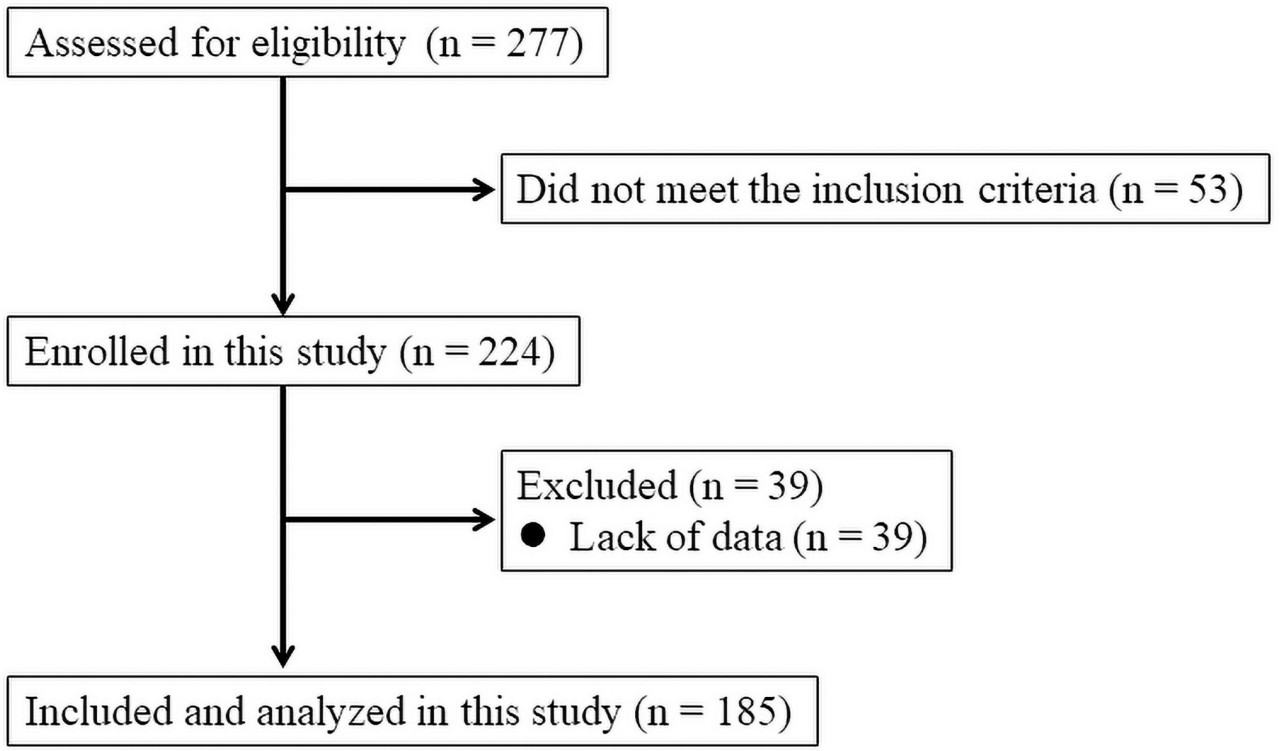

**Fig 1. Patients flow chart.**

Hospital (Niigata, Japan; approval number, H29-11) and conformed to the provisions of the Declaration of Helsinki (as revised in Tokyo in 2004).

### Patients' baseline characteristics and clinical laboratory measurements

The patients' baseline characteristics and clinical data were collected from their medical charts, while data on the primary disease leading to the need for dialysis and the coexistence of comorbid cardiovascular or cerebrovascular diseases were extracted from their medical records. BP and heart rate were measured before HD with the patients in the supine position. Blood samples were obtained at ambient temperatures from the HD access points, such as arteriovenous fistulas, arteriovenous grafts, and HD catheters, from all the patients before HD. Peripheral blood counts and biochemical parameters were evaluated in all the patients.

### Monitoring of hepatic oxygenation

Hepatic $rSO_2$, a marker of hepatic oxygenation, was monitored using the INVOS 5100c saturation monitor (Covidien Japan, Tokyo, Japan) described previously [10]. Briefly, this instrument uses a light-emitting diode that transmits near-infrared light at two wavelengths (735 and 810 nm) and two silicon photodiodes that act as light detectors that measure oxygenated and deoxygenated hemoglobin (Hb). The ratio of the signal strengths of the oxygenated Hb and the total Hb (oxygenated Hb + deoxygenated Hb) was calculated, and the corresponding percentage was recorded as a single numerical value that represented the $rSO_2$ [5,6]. All data obtained using this instrument were immediately and automatically stored. Furthermore, the light paths leading from the emitter to the different detectors share a common part: the

30-mm detector assesses superficial tissue, whereas the 40-mm detector assesses deep tissue. By analyzing the differential signals collected by the two detectors, cerebral $rSO_2$ values in the deep tissue were obtained from a distance of 20–30 mm from the body surface [13,14]. These measurements were performed at 6-s intervals.

Prior to HD, the participants rested in the supine position for at least 10 min to reduce the influence of postural changes on $rSO_2$. An $rSO_2$ measuring sensor was attached to each patient's right intercostal area above the liver to measure the resting-state $rSO_2$ levels. The right intercostal area just above the liver was identified on ultrasonography before HD. The $rSO_2$ level was measured for 5 min before HD, and the mean $rSO_2$ value was calculated as a measure of cerebral oxygenation in each patient.

## Calculation of colloid osmotic pressure

Intravascular colloid osmotic pressure (COP) is important for maintaining the systemic tissue microcirculation. Therefore, to examine the influence of the COP on hepatic $rSO_2$, the COP before HD was calculated using the equation below (a specialized method for HD patients) [15]:

$$COP \text{ (mmHg)} = -7.91 + 5.64 \times \text{serum albumin (g/dL)} + 3.00 \times [\text{total protein (g/dL)} - \text{serum albumin (g/dL)}]$$

## Statistical analysis

Data are expressed as mean ± standard deviation or median and interquartile range. The normality of the hepatic $rSO_2$ in each of the HD patient and healthy control groups was assessed using the Shapiro-Wilk test. The results of hepatic $rSO_2$ in each group were not significant (hepatic $rSO_2$ in HD patients, p = 0.185; healthy control, p = 0.236). Therefore, hepatic $rSO_2$ distribution in each group was confirmed to be normal. The differences in hepatic $rSO_2$ levels between healthy controls and patients undergoing HD were evaluated using the unpaired Student's t-test. Variables with P- values < 0.05 in a simple linear regression analysis were included in the multivariable linear regression analysis to identify factors affecting hepatic $rSO_2$ in patients undergoing HD. HD duration and C-reactive protein (CRP) levels were transformed using the natural logarithm (Ln) in the regression analyses because they had a skewed distribution. All analyses were performed using IBM SPSS Statistics for Windows, version 26.0 (IBM, Armonk, NY, USA). Statistical significance was set at P < 0.05.

## Results

The patients' general characteristics and the correlations between hepatic $rSO_2$ and clinical parameters are summarized in Table 1. The mean hepatic $rSO_2$ was significantly lower in patients undergoing HD than in healthy controls (56.4 ± 14.9% vs. 76.2 ± 9.6%, p < 0.001; Fig 2). Furthermore, hepatic $rSO_2$ was significantly positively correlated with body mass index (BMI), mean BP, interdialytic weight gain, Hb levels, the serum creatinine concentration, the serum albumin concentration, the colloid osmotic pressure, and the use of renin-angiotensin-aldosterone system (RAS) inhibitors and calcium channel blockers and negatively correlated with age, a history of cardiovascular disease, the aspartate aminotransferase level, and Ln-CRP levels.

The results of the multivariable linear regression analysis are presented in Tables 2 and 3. For Model 1, age, BMI, the mean BP, a history of cardiovascular disease, Hb levels, the serum

**Table 1. Characteristics of patients undergoing hemodialysis and correlations between hepatic rSO$_2$ and clinical parameters.**

| | | Simple linear regression vs. hepatic rSO$_2$ | |
|---|---|---|---|
| **Patient characteristics** | | **r** | **P** |
| Number of patients, n | 185 | | |
| Hepatic rSO$_2$ (%) | 56.4 ± 14.9 | | |
| Men/women, n | 132/53 | -0.137 | 0.064 |
| Age, years | 68.3 ± 10.9 | -0.209 | 0.004* |
| Body mass index, kg/m$^2$ | 22.2 ± 3.4 | 0.470 | < 0.001* |
| Mean BP, mmHg | 97.6 ± 15.9 | 0.391 | < 0.001* |
| Heart rate, beats/min | 73.8 ± 15.4 | 0.019 | 0.792 |
| O$_2$ saturation, % | 95.1 ± 3.3 | -0.077 | 0.298 |
| Causes of chronic renal failure | | | |
| Diabetes mellitus, n (%) | 76 (41) | 0.071 | 0.334 |
| Chronic glomerulonephritis, n (%) | 43 (23) | 0.033 | 0.654 |
| Other, n (%) | 66 (36) | | |
| Comorbidities | | | |
| Cardiovascular disease, n (%) | 65 (35) | -0.249 | 0.001* |
| Cerebrovascular disease, n (%) | 35 (19) | -0.048 | 0.513 |
| HD-associated parameters | | | |
| HD duration, years, median (interquartile range) | 0.7 (0.1–6.0) | -0.103 | 0.164 |
| HD time, h | 3.7 ± 0.6 | 0.018 | 0.812 |
| Interdialytic weight gain, kg | 1.7 ± 1.0 | 0.178 | 0.016* |
| Laboratory findings | | | |
| Hemoglobin, g/dL | 9.9 ± 1.6 | 0.470 | < 0.001* |
| BUN, mg/dL | 55.8 ± 18.1 | 0.137 | 0.064 |
| Serum creatinine, mg/dL | 8.4 ± 2.4 | 0.369 | < 0.001* |
| Total bilirubin, mg/dL | 0.4 ± 0.4 | 0.004 | 0.958 |
| AST, IU/L | 17 ± 12 | -0.185 | 0.012* |
| ALT, IU/L | 13 ± 10 | 0.006 | 0.936 |
| LDH, IU/L | 222 ± 94 | -0.076 | 0.302 |
| CRP, mg/dL, median (interquartile range) | 0.3 (0.1–1.2) | -0.285 | < 0.001* |
| Serum albumin, g/dL | 3.2 ± 0.5 | 0.371 | < 0.001* |
| Colloid osmotic pressure, mmHg | 18.7 ± 2.9 | 0.330 | < 0.001* |
| Medication | | | |
| RAS inhibitors, n (%) | 93 (50) | 0.201 | 0.006* |
| Calcium channel blockers, n (%) | 114 (62) | 0.213 | 0.004* |

Values are shown as counts (with or without percentages) or as mean ± standard deviation unless noted otherwise.

*Statistically significant.

ALT, alanine transaminase; AST, aspartate aminotransferase; BP, blood pressure; BUN, blood urea nitrogen; CRP, C-reactive protein; HD, hemodialysis; LDH, lactate dehydrogenase; O$_2$, oxygen; RAS, renin-angiotensin-aldosterone system; rSO$_2$, regional oxygen saturation.

creatinine concentration, the serum albumin concentration, the aspartate aminotransferase level, Ln-CRP levels, and the use of RAS inhibitors and calcium channel blockers, as variables with P values < 0.05, were included in the multivariable linear regression analysis. As shown in Table 2, hepatic rSO$_2$ was independently associated with BMI (standardized coefficient: 0.294), Hb levels (standardized coefficient: 0.294), a history of cardiovascular disease (standardized coefficient: -0.157), the mean BP (standardized coefficient: 0.154), and the serum albumin concentration (standardized coefficient: 0.150). The COP value was included in place

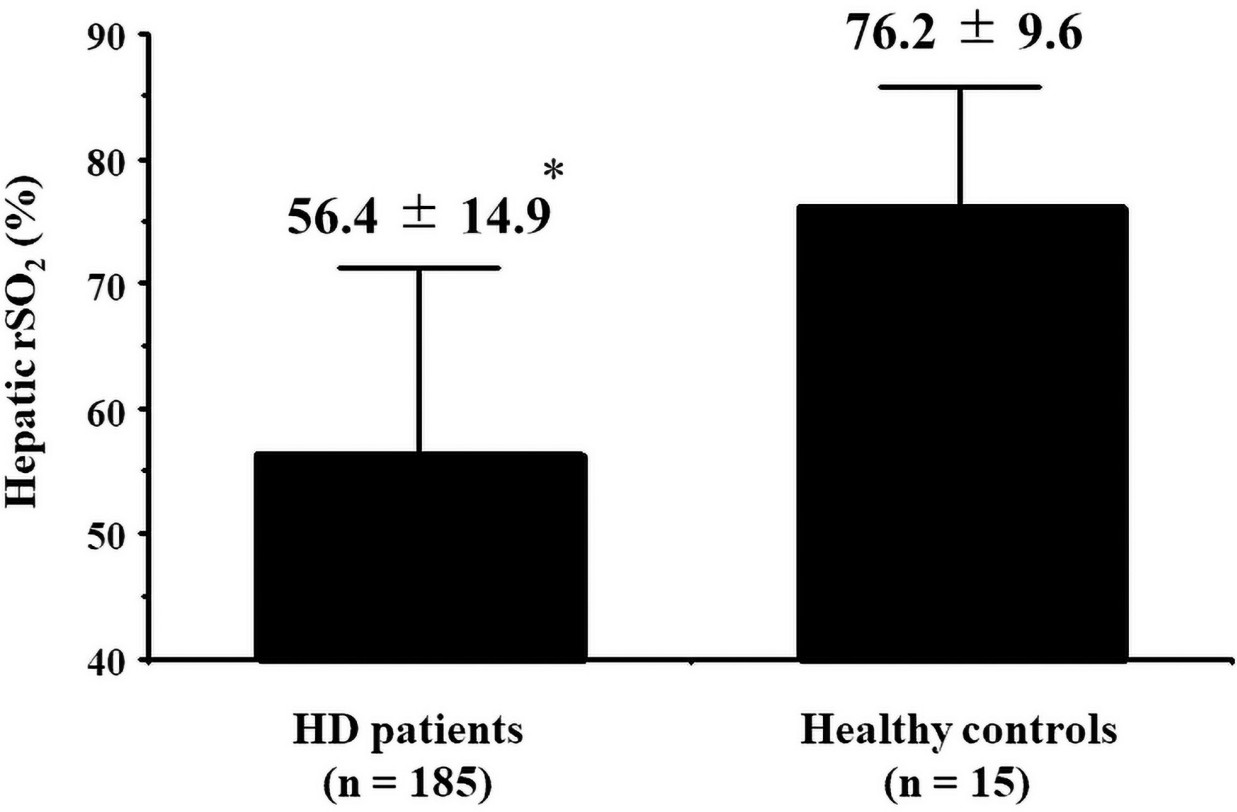

**Fig 2. Comparison of hepatic rSO₂ between patients undergoing HD and healthy controls.** Abbreviations: HD, hemodialysis; rSO2, regional saturation of oxygen. *p < 0.001 vs. healthy controls.

**Table 2. Multivariable linear regression analysis in Model 1 using serum albumin: Factors independently associated with hepatic rSO₂ in patients undergoing HD.**

| Variables | Coefficient | 95% CI | Standardized coefficient | P |
|---|---|---|---|---|
| Age | | | -0.022 | 0.713 |
| Body mass index | 1.667 | 1.175–3.770 | 0.294 | < 0.001* |
| Mean BP | 0.145 | 0.035–0.255 | 0.154 | 0.010* |
| History of cardiovascular disease | -4.906 | -8.317 to -1.494 | -0.157 | 0.005* |
| Interdialytic weight gain | | | 0.054 | 0.351 |
| Hemoglobin | 2.668 | 1.566–3.770 | 0.294 | < 0.001* |
| Serum creatinine | | | 0.105 | 0.095 |
| Serum albumin | 4.114 | 0.017–0.760 | 0.150 | 0.017* |
| AST | | | -0.040 | 0.485 |
| Ln-CRP | | | -0.056 | 0.413 |
| Use of RAS inhibitors | | | 0.044 | 0.437 |
| Use of calcium channel blockers | | | 0.086 | 0.133 |

*Statistically significant.

AST, aspartate aminotransferase; BP, blood pressure; CI, confidence interval; CRP, C-reactive protein; HD, hemodialysis; Ln, natural logarithm; RAS, renin-angiotensin-aldosterone system; rSO₂, regional oxygen saturation.

**Table 3. Multivariable linear regression analysis in Model 2 using colloid osmotic pressure: Factors independently associated with hepatic rSO$_2$ in patients undergoing HD.**

| Variables | Coefficient | 95% CI | Standardized coefficient | P |
|---|---|---|---|---|
| Age | | | -0.028 | 0.645 |
| Body mass index | 1.671 | 1.179–2.164 | 0.377 | < 0.001* |
| Mean BP | 0.151 | 0.041–0.261 | 0.161 | 0.007* |
| History of cardiovascular disease | -5.183 | -8.615 to -1.751 | -0.166 | 0.003* |
| Interdialytic weight gain | | | 0.044 | 0.446 |
| Hemoglobin | 2.726 | 1.625–3.826 | 0.301 | < 0.001* |
| Serum creatinine | | | 0.111 | 0.075 |
| Colloid osmotic pressure | 0.686 | 0.029–0.072 | 0.134 | 0.029* |
| AST | | | -0.048 | 0.408 |
| Ln-CRP | | | -0.079 | 0.221 |
| Use of RAS inhibitors | | | 0.043 | 0.454 |
| Use of calcium channel blockers | | | 0.088 | 0.128 |

*Statistically significant.

AST, aspartate aminotransferase; BP, blood pressure; CI, confidence interval; CRP, C-reactive protein; HD, hemodialysis; Ln, natural logarithm; RAS, renin-angiotensin-aldosterone system; rSO$_2$, regional oxygen saturation.

of serum albumin concentration in Model 2 to avoid collinearity with the serum albumin concentration. As a result, the COP (standardized coefficient: 0.134) was also identified as factors affecting hepatic rSO$_2$ in addition to BMI (standardized coefficient: 0.377), Hb levels (standardized coefficient: 0.301), a history of cardiovascular disease (standardized coefficient: -0.166), and the mean BP (standardized coefficient: 0.161) (Table 3).

## Discussion

In this study, a significant decrease in hepatic rSO$_2$ was confirmed in patients undergoing HD compared to healthy controls. In addition, hepatic rSO$_2$ levels were independently associated with BMI, Hb levels, a history of cardiovascular disease, the mean BP, the serum albumin concentration, and the COP on the multivariable linear regression analysis. We confirmed that each adjusted R$^2$ reflects the fitness of model; those in Models 1 and 2 were 0.455 and 0.452, respectively. Therefore, these results are in the range of moderately good results for an exploratory study.

A difference in hepatic rSO$_2$ between patients undergoing HD and healthy controls was also confirmed in this study. Thus far, significant decreases in cerebral oxygenation in patients undergoing HD compared with healthy controls have been reported [16–18] that might be explained by renal anemia and a decrease in the serum albumin concentration [17,18]. Therefore, the results of this study are consistent with those of previous reports of tissue oxygenation in patients undergoing HD, although this result was inconclusive because the ages and sexes were not completely matched between the groups.

In the present study, BMI was the most significantly positive factor associated with hepatic rSO$_2$. In patients undergoing HD, in contrast to the general population, a higher BMI was reportedly associated with better survival, a phenomenon referred to as reverse epidemiology [19,20]. Serum leptin is reportedly associated with reverse epidemiology in patients undergoing HD [21–23]. Serum leptin has emerged as a potential risk factor for cardiovascular disease due to its proinflammatory, proatherogenic, and prothrombotic effects, which promote endothelial dysfunction [24]. However, in patients undergoing HD, the serum leptin concentration

was significantly and positively correlated with BMI [25] and negatively associated with serum CRP [26] and Malnutrition-Inflammation Score [22]. Furthermore, there was a significant positive association between the serum leptin concentration and cardiac function [23]. Therefore, according to the increase in BMI, a serum leptin concentration increase might be expected and positively influence the hepato-splanchnic circulation by improving cardiac function. However, serum leptin concentrations were not measured in the present study. Therefore, we cannot directly comment on the effect of serum leptin on the association between hepatic oxygenation and BMI.

Hb plays an important role in carrying oxygen to the systemic tissues, including the liver; therefore, systemic oxygenation is believed to be associated with the Hb level. The hepatic $rSO_2$ of patients with severe anemia undergoing HD was low, but it improved significantly in response to the increase in Hb following an intradialytic blood transfusion (hepatic $rSO_2$, from $46.7 \pm 1.7\%$ to $55.4 \pm 2.0\%$ before versus after intradialytic blood transfusion), and changes in hepatic $rSO_2$ were positively and significantly associated with the transfusion-induced Hb increase [10]. In this study, similar to the association between hepatic $rSO_2$ and BMI, the Hb level had a significant positive effect on hepatic oxygenation. Therefore, Hb levels targeted in the clinical setting of HD therapy, which was considered the upper limit for the appropriate management of renal anemia [27–29], would contribute to maintaining and/or improving the hepato-splanchnic circulation and oxygenation of patients undergoing HD. In addition, the hepatic artery buffer response, which is increased in the hepatic artery flow to compensate for the reduction of portal vein flow, plays an important role in maintaining hepatic circulation under various clinical conditions [30,31]. However, beyond the protective effect of the hepatic artery buffer response, prolonged decreases in the cardiac output and mean BP induced by cardiac tamponade are reportedly associated with a decreased blood flow in the hepatic artery [32], which would lead to a decrease in hepatic oxygenation. Furthermore, hepatic $rSO_2$ values were significantly correlated with cardiac output, systolic BP, and diastolic BP [8]. In this study, the mean BP showed a significantly positive influence on hepatic $rSO_2$, whereas a history of cardiovascular disease was negatively associated with hepatic $rSO_2$. Although this study did not confirm the influence of a history of cardiovascular disease on cardiac function, it might be associated with a decrease in cardiac output in patients undergoing HD. Therefore, the result of a negative association with a history of cardiovascular disease and a positive association with mean BP on hepatic oxygenation might be consistent with those of previous reports [8,32]. Serum albumin represents an oncotic and non-oncotic effect, including the formation of COP and anti-oxidant and anti-inflammatory properties [33–35]. In particular, an increase in the serum albumin concentration reportedly improved cerebral oxygenation by improving the cerebral microcirculation associated with the oncotic pressure effect [18]. In this study, the serum albumin concentration and COP were analyzed in separate models to determine whether the association between hepatic oxygenation and the serum albumin concentration was due to the effect of the oncotic pressure effect. The results of this study showed that serum albumin concentration and COP were significantly and positively correlated with hepatic $rSO_2$. Therefore, the effect of serum albumin on hepatic oxygenation might be at least partially associated with the oncotic pressure effect influenced by the serum albumin concentration in patients undergoing HD.

The present study had several limitations. First, its sample size was relatively small. Second, the history of cardiovascular diseases was extracted from the patients' medical records, and cardiac functions was not always confirmed using echocardiography. However, the assessment of cardiac function via echocardiography would be essential to clarify the association between the clinical history and stages of cardiovascular disease, and the status of hepato-splanchnic circulation, including hepatic $rSO_2$. Therefore, further evaluation of this study, including the

assessment of cardiac function via echocardiography as a confounding factor, would be required. Third, the residual renal function and diuresis play an important role in the body fluid management of HD patients via the prevention of interdialytic weight gain, which may lead to the reduction in the need for aggressive ultrafiltration and the stability in hepato-splanchnic circulation. Furthermore, because of the increases in urinary volume and sodium excretion associated with the usage of furosemide even in HD patients [36], it is important to check the usage of diuretics. However, residual diuresis volume was not measured, and the diuretics usage was not quantified in this study. Therefore, the association between hepatic $rSO_2$ and the residual renal function, including diuresis, remains unclear. Fourth, the type of dialysis modality and differences in dialyzer membrane may influence the hepatic $rSO_2$ values due to the increase of albumin loss into the dialysate and decrease in serum albumin concentration [37,38]. Information regarding the dialyzer membrane was not collected in this study, although the only method of dialysis was HD for all patients included in this study. The differences in dialysis modalities and dialysis membranes may affect the hepatic oxygenation; hence, further studies are needed. In addition, regarding the patients' characteristics in this study, the mean age in this study was 68.3 ± 10.9 years, while that in Japanese dialysis patients was reportedly 68.75 years [39]. Furthermore, 41% and 23% of the causes of chronic renal failure in this study were diabetes mellitus and chronic glomerulonephritis, respectively, while these two causes were reported as 39.0% and 26.8% in Japanese patients, respectively [39]. Therefore, the age and causes of chronic renal failure in this study could be considered similar to those in Japanese dialysis patients. However, the median HD duration in this study was 0.7 years, while the mean dialysis period in Japanese patients was 6.82 years [39]. Based on these characteristics in this study, the shortness of HD duration in this study compared with those in Japanese dialysis patients may have affected the hepatic oxygenation status. Finally, since the present study had a cross-sectional design, it was impossible to assess the directionality of the association between hepatic oxygenation and clinical factors. Therefore, future studies are required to comprehensively investigate the association between hepatic $rSO_2$ and clinical parameters.

## Conclusion

The hepatic oxygenation before HD might be positively associated with the Hb level, mean BP, serum albumin concentration, and COP as well as BMI and a history of cardiovascular disease. Further prospective studies are needed to clarify whether changes in these parameters, including during HD, affect the hepato-splanchnic circulation and oxygenation of patients undergoing HD.

## Supporting information

**S1 Dataset.**
(XLSX)

## Acknowledgments

We thank the study participants and dialysis staff of Minami-Uonuma City Hospital and our hospital.

## Author Contributions

**Conceptualization:** Yuichiro Ueda, Susumu Ookawara, Kiyonori Ito.

**Data curation:** Yuichiro Ueda, Susumu Ookawara.

**Formal analysis:** Susumu Ookawara, Kiyonori Ito, Yusuke Sasabuchi.

**Funding acquisition:** Susumu Ookawara.

**Investigation:** Yuichiro Ueda, Susumu Ookawara, Kiyonori Ito, Hideyuki Hayasaka, Masaya Kofuji, Takayuki Uchida, Sojiro Imai, Satoshi Kiryu, Saori Minato, Haruhisa Miyazawa, Hidenori Sanayama, Keiji Hirai.

**Methodology:** Yuichiro Ueda, Susumu Ookawara, Kiyonori Ito.

**Project administration:** Susumu Ookawara.

**Supervision:** Kaoru Tabei, Yoshiyuki Morishita.

**Validation:** Yuichiro Ueda, Susumu Ookawara, Kiyonori Ito.

**Writing – original draft:** Yuichiro Ueda, Susumu Ookawara, Kiyonori Ito.

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
