## [Decision Letter · Decision Letter 0]

18 Jun 2021

PONE-D-21-15838

Association between hepatic oxygenation on near-infrared spectroscopy and clinical factors in patients undergoing hemodialysis

PLOS ONE

Dear Dr. Ookawara,

Thank you for submitting your manuscript to PLOS ONE. After careful consideration, we feel that it has merit but does not fully meet PLOS ONE’s publication criteria as it currently stands. Therefore, we invite you to submit a revised version of the manuscript that addresses the points raised during the review process.

We look forward to receiving your revised manuscript.

Kind regards,

Abduzhappar Gaipov

Academic Editor

PLOS ONE

Journal Requirements:

2. In your Methods section, please provide additional information about the participant recruitment method and the demographic details of your participants. Please ensure you have provided sufficient details to replicate the analyses such as: 

a) the recruitment date range (month and year), 

b) a description of any inclusion/exclusion criteria that were applied to participant recruitment, 

c) a statement as to whether your sample can be considered representative of a larger population, and 

d) a description of how participants were recruited.

Reviewers' comments:

Reviewer's Responses to Questions

**Comments to the Author**

1. Is the manuscript technically sound, and do the data support the conclusions?

Reviewer #1: Yes

Reviewer #2: No

Reviewer #3: Yes

2. Has the statistical analysis been performed appropriately and rigorously? 

Reviewer #1: Yes

Reviewer #2: No

Reviewer #3: Yes

3. Have the authors made all data underlying the findings in their manuscript fully available?

Reviewer #1: Yes

Reviewer #2: No

Reviewer #3: Yes

4. Is the manuscript presented in an intelligible fashion and written in standard English?

Reviewer #1: Yes

Reviewer #2: Yes

Reviewer #3: Yes

5. Review Comments to the Author

Reviewer #1: Just a few remarks:

page 13 - table - "O2 Saturation"; the given CIs would add up to more than 100%, this needs to be corrected.

This study is a very nice approach to the fluid assessment and fluid control problem in HD patients; nevertheless this study should include the residual kidney function of the patients, meaning the residual diuresis which may significantly contribute to the total blood volume stability.

This should be adressed.

Another weak point, is the lack of thourough information about the stage of cardivascular disease the patients are suffering from. Assessment via echocardiography or classification seems to be pivotal for further valuation of this data.

Reviewer #2: PONE-D-21-15838: statistical review

SUMMARY. This is a cross-sectional study on the associations between a battery of covariates and Hepatic rSO2 in patients undergoing hemodialysis (HD). The core statistical analysis relies on two multivariate regression analyses displayed by Tables 2 and 3. I list below some serious concerns I have about the methods.

MAJOR ISSUES

1. Figure 1 shows the distribution of Hepatic rSO2 in HD patients and healthy subjects. It seems that both distributions are skew (nonnormal). A Student t for testing the differences in hepatic rSO2 levels between healthy controls and HD patients is therefore not fully appropriate here. I’d suggest a nonparametric approach.

2. Linear regression models require normality of the response variable, Hepatic rSO2. Because figure 1 indicates that the response distribution is skew, the analysis must be revised. I see two options here. Either the response variable is transformed to recast normality (it is possible that a simple logarithmic transformation could do the job); or, a skew normal regression model must be estimated. Skew normal regression models are available in popular statistical software such as R.

3. It seems that multivariate regressions involve only HD patients (at least, this is what the captions of Tables 2 and 3 say). Why? Shouldn’t the control individuals be included too? I guess it would be important to check whether the significant associations observed in HD subjects are still significant for healthy subjects. For example, BMI is a significant predictor of Hepatic rSO2 in HD patients. Is this association still significant for healthy subjects? Issues such as this one could be easily investigated by including interaction terms in the linear regression models.

4. Model checking is overlooked. Could at least the authors provide some evidence of the goodness of fit of the models? I’m asking this because the p-values are not very small. On one side, this could be due to the small sample size but, on the other side, it could indicate a poor fit of the estimated models.

5. 39 patients were excluded from the analysis due to a lack of data. Could the authors provide evidence that the data are missing at random? If the data are not missing at random (i.e. the probability of a missing value depends on the unobserved value), then the subsequent regression analysis could be severely biased.

Reviewer #3: In this interesting study analyzing the neglected hepatic blood supply and oxygenation authors show at basal level a decrease of hepatic regional oxygen saturation which was found in multivariable analyses to be association with BMI, Hb level, an history of cardiovascular disease, mean blood pressure, serum albumin and colloid oncotic pressure.

I just suggest the need to the authors ro analyze the possible influence of hemodialysis technic on hepatic regional oxygen saturation eg : traditional hemodialysis with polysulfone membranes, HDx with THERANOVA, Hemodiafiltration, protein leaking hemodialysis with PMMA membrane and adsorption dialysis with AN69 membrane.

6. PLOS authors have the option to publish the peer review history of their article (what does this mean?). If published, this will include your full peer review and any attached files.

Reviewer #1: No

Reviewer #2: No

Reviewer #3: No

---

## [Author Response · Author response to Decision Letter 0]

13 Jul 2021

Response to the editors’ and reviewers’ comments

We appreciate your careful review of our manuscript and thank you for the opportunity to submit our revised manuscript and point-by-point response to the editors’ and reviewers’ comments. We hope that we have satisfactorily addressed all issues raised by the academic editor and reviewers.

Journal Requirements

Comment 1:

Response 1:

Thank you for your comment. As the editor advised, we revised the manuscript according to the PLoS ONE style requirements.

Comment 2:

In your Methods section, please provide additional information about the participant recruitment method and the demographic details of your participants. Please ensure you have provided sufficient details to replicate the analyses such as: 

 a) the recruitment date range (month and year), 

 b) a description of any inclusion/exclusion criteria that were applied to participant recruitment, 

 c) a statement as to whether your sample can be considered representative of a larger population

d) a description of how participants were recruited.

Response 2:

a) We quantified the recruitment date range in the “Materials and Methods” section in the revised manuscript as follows:

Page 7, Lines 13-15:

“Fig. 1 shows a flow chart of patient enrollment and analysis. Of the 277 patients screened, 224 met the inclusion criteria and were enrolled between August 1, 2013 and December 31, 2019.”

b) We quantified the inclusion and exclusion criteria in the revised manuscript as follows:

Page 7, Lines 5-12:

“This study was performed at two facilities, including our hospital. Patients who met the following criteria were enrolled: (i) age > 20 years; (ii) end-stage renal disease managed with HD; (iii) started HD at least one month before the study; (iv) tissue thickness ≤ 20 mm from the skin to the surface of the liver in the right intercostal area as measured by ultrasonography; and (v) hepatic rSO2 data collected using an INVOS 5100c oxygen saturation monitor. The exclusion criteria were coexisting major diseases, including congestive heart failure or neurological disorders, such as severe cerebrovascular disease and cognitive impairment.”

c) Thank you for your insightful comment. As the editor pointed out, it is important to note whether characteristics of HD patients included in this study reflect those in a larger HD population. The mean age in this study was 68.3 ± 10.9 years, while that in Japanese dialysis patients was reportedly 68.75 years (Ref 39). Furthermore, 41% and 23% of the causes of chronic renal failure in this study were diabetes mellitus and chronic glomerulonephritis, respectively, while these two causes were reported as 39.0% and 26.8% in Japanese patients, respectively. (Ref 39). Therefore, the age and causes of chronic renal failure in this study could be considered similar to those in Japanese dialysis patients. However, the median HD duration in this study was 0.7 years, while the mean dialysis period in Japanese patients was 6.82 years (Ref 39). The shortness of HD duration in this study compared with those in Japanese dialysis patients may have affected the hepatic oxygenation status. We added these to the “Limitation” section in the revised manuscript as follows:

Page 25, Lines 5-15:

“In addition, regarding the patients’ characteristics in this study, the mean age in this study was 68.3 ± 10.9 years, while that in Japanese dialysis patients was reportedly 68.75 years [39]. Furthermore, 41% and 23% of the causes of chronic renal failure in this study were diabetes mellitus and chronic glomerulonephritis, respectively, while these two causes were reported as 39.0% and 26.8% in Japanese patients, respectively [39]. Therefore, the age and causes of chronic renal failure in this study could be considered similar to those in Japanese dialysis patients. However, the median HD duration in this study was 0.7 years, while the mean dialysis period in Japanese patients was 6.82 years [39]. Based on these characteristics in this study, the shortness of HD duration in this study compared with those in Japanese dialysis patients may have affected the hepatic oxygenation status.”

Reference 39

Nitta K, Goto S, Masakane I, Hanafusa N, Taniguchi M, Hasegawa T, et al. Annual dialysis report for 2018, JSDT Renal Data Registry: survey methods, facility data, incidence, prevalence, and mortality. Renal Replacement Therapy. 2020;6: 41.

d) As the editor advised, we added the “Patients’ flow chart” to Fig. 1.

Comment 3:

Upon re-submitting your revised manuscript, please upload your study’s minimal underlying data set as either Supporting Information files or to a stable, public repository and include the relevant URLs, DOIs, or accession numbers within your revised cover letter.

Responses 3:

As the editor advised, we uploaded our study’s data set as a supporting file in our resubmission.

We hope that our revised manuscript is now suitable for publication in your highly esteemed journal.

Reviewer 1

Comment 1:

page 13 - table - "O2 Saturation"; the given CIs would add up to more than 100%, this needs to be corrected.

Response 1:

As the reviewer pointed out, we made a mistake in describing O2 saturation values. We have corrected the description of O2 saturation values and the results of simple linear regression between the O2 saturation and hepatic rSO2 in the revised manuscript.

Comment 2:

This study is a very nice approach to the fluid assessment and fluid control problem in HD patients; nevertheless this study should include the residual kidney function of the patients, meaning the residual diuresis which may significantly contribute to the total blood volume stability. This should be addressed.

Response 2:

Thank you for your insightful comment. As the reviewer suggested, the residual renal function and diuresis play an important role in the body fluid management of HD patients via the prevention of interdialytic weight gain, which may lead to the reduction in the need for aggressive ultrafiltration and the stability in hepato-splanchnic circulation. Furthermore, because of increased urinary volume and sodium excretion associated with furosemide use in HD patients (Clin Exp Nephrol 2011;15: 554-559), it is important to check the use of diuretics. However, residual diuresis volume was not measured in this study; thus, the diuretics usage was not quantified. We added the following in the “Limitations” section in the revised manuscript:

Page 24, Lines 8-17:

“Third, the residual renal function and diuresis play an important role in the body fluid management of HD patients via the prevention of interdialytic weight gain, which may lead to the reduction in the need for aggressive ultrafiltration and the stability in hepato-splanchnic circulation. Furthermore, because of the increases in urinary volume and sodium excretion associated with the usage of furosemide even in HD patients [36], it is important to check the usage of diuretics. However, residual diuresis volume was not measured, and the diuretics usage was not quantified in this study. Therefore, the association between hepatic rSO2 and the residual renal function, including diuresis, remains unclear.”

Reference 36

Lemes HP, Araujo S, Nascimento D, Cunha D, Garcia C, Queiroz V, et al. Use of small doses of furosemide in chronic kidney disease patients with residual renal function undergoing hemodialysis. Clin Exp Nephrol. 2011;15: 554-559.

Comment 3:

Another weak point, is the lack of thourough information about the stage of cardivascular disease the patients are suffering from. Assessment via echocardiography or classification seems to be pivotal for further evaluation of this data.

Response 3:

We completely agree with the reviewer’s comment. As suggested, the assessment of cardiac function via echocardiography would be essential to clarify the association between the clinical history and stages of cardiovascular disease, and the status of hepato-splanchnic circulation, including hepatic rSO2. We rewrote these in the “Limitations” section in the revised manuscript as follows:

Page 24, Lines 4-8:

“However, the assessment of cardiac function via echocardiography would be essential to clarify the association between the clinical history and stages of cardiovascular disease, and the status of hepato-splanchnic circulation, including hepatic rSO2. Therefore, further evaluation of this study, including the assessment of cardiac function via echocardiography as a confounding factor, would be required.”

Reviewer 2

Comment 1:

Figure 1 shows the distribution of Hepatic rSO2 in HD patients and healthy subjects. It seems that both distributions are skew (nonnormal). A Student t for testing the differences in hepatic rSO2 levels between healthy controls and HD patients is therefore not fully appropriate here. I’d suggest a nonparametric approach.

Response 1:

Thank you for your comments. In this study, the normality of the hepatic rSO2 in each of the HD patient and healthy control groups was assessed using the Shapiro-Wilk test. The results of hepatic rSO2 in each group were not significant (hepatic rSO2 in HD patients, p = 0.185; healthy control, p = 0.236). Therefore, the hepatic rSO2 distribution in each group was confirmed to be normal. We added these in the “Statistical analysis” section as follows:

Page 11, Lines 12-15:

“The normality of the hepatic rSO2 in each of the HD patient and healthy control groups was assessed using the Shapiro-Wilk test. The results of hepatic rSO2 in each group were not significant (hepatic rSO2 in HD patients, p = 0.185; healthy control, p = 0.236). Therefore, hepatic rSO2 distribution in each group was confirmed to be normal.”

Comment 2:

Linear regression models require normality of the response variable, Hepatic rSO2. Because figure 1 indicates that the response distribution is skew, the analysis must be revised. I see two options here. Either the response variable is transformed to recast normality (it is possible that a simple logarithmic transformation could do the job); or, a skew normal regression model must be estimated. Skew normal regression models are available in popular statistical software such as R.

Response 2:

As mentioned above, we confirmed the normality of the hepatic rSO2, as a response variable in this study, using the Shapiro-Wilk test for the HD patients. Therefore, we did not change the analyses for examining the association between hepatic rSO2 and clinical parameters in our revised manuscript.

Comment 3:

It seems that multivariate regressions involve only HD patients (at least, this is what the captions of Tables 2 and 3 say). Why? Shouldn’t the control individuals be included too? I guess it would be important to check whether the significant associations observed in HD subjects are still significant for healthy subjects. For example, BMI is a significant predictor of Hepatic rSO2 in HD patients. Is this association still significant for healthy subjects? Issues such as this one could be easily investigated by including interaction terms in the linear regression models.

Response 3:

We agree with the reviewer’s comment. There were no medical problems in the healthy control group included in this study as they did not take any medications. However, we could only provide data on their age, sex, and hepatic rSO2 values because we could not obtain other clinical parameters, including their biochemical parameters and arterial blood pressure. Furthermore, this study aimed to elucidate the clinical factors influencing the hepatic rSO2 in patients undergoing HD. Therefore, we could not include healthy individuals in the analysis.

Comment 4:

Model checking is overlooked. Could at least the authors provide some evidence of the goodness of fit of the models? I’m asking this because the p-values are not very small. On one side, this could be due to the small sample size but, on the other side, it could indicate a poor fit of the estimated models.

Responses 4:

Thank you for your thoughtful suggestions. We now understand your suggestion and confirmed each adjusted R2, to reflect the fitness of model, in Models 1 and 2 in this study. Those in Models 1 and 2 were 0.455 and 0.452, respectively. Therefore, these results are in the range of moderately good results for an exploratory study. We added the following in the “Discussion” section in the revised manuscript as follows:

Page 20, Lines 7-9:

“We confirmed that each adjusted R2 reflects the fitness of model; those in Models 1 and 2 were 0.455 and 0.452, respectively. Therefore, these results are in the range of moderately good results for an exploratory study.”

Comment 5:

39 patients were excluded from the analysis due to a lack of data. Could the authors provide evidence that the data are missing at random? If the data are not missing at random (i.e. the probability of a missing value depends on the unobserved value), then the subsequent regression analysis could be severely biased.

Response 5:

We agree with your suggestion and understand your comment. It was difficult to prove that the data in 39 patients excluded in this study were missing at random. However, hepatic rSO2 values, as a response variable, were 56.9 ± 16.1% in these excluded patients; there were no differences in hepatic rSO2 between the included HD patients (56.4 ± 14.9%) and excluded patients (p = 0.817). Furthermore, the normality of the hepatic rSO2 in the 39 excluded patients was assessed using the Shapiro-Wilk test. The result of hepatic rSO2 in these patients was not significant (p = 0.603); the hepatic rSO2 distribution in these patients was confirmed to be normal. Therefore, the hepatic rSO2 in HD patients included in this study might not contain a bias that is independent of the 39 excluded patients.

Reviewer 3

Comment 1:

I just suggest the need to the authors to analyze the possible influence of hemodialysis technic on hepatic regional oxygen saturation eg : traditional hemodialysis with polysulfone membranes, HDx with THERANOVA, Hemodiafiltration, protein leaking hemodialysis with PMMA membrane and adsorption dialysis with AN69 membrane.

Response 1:

Thank you for your comment. As the reviewer suggested, the kinds of dialysis modality and differences in dialyzer membrane may influence the hepatic rSO2 values due to the increase of albumin loss into the dialysate and decrease in serum albumin concentration (Ref 37.38). The information regarding the dialyzer membrane was not collected in this study, although the only method of dialysis was HD for all the patients included. The differences in dialysis modalities and dialysis membranes may affect the hepatic oxygenation; hence, further studies are needed in the future. We added these in the “Limitations” section in the revised manuscript as follows:

Page 24, Line 17-Page 25, Line 5:

“Fourth, the type of dialysis modality and differences in dialyzer membrane may influence the hepatic rSO2 values due to the increase of albumin loss into the dialysate and decrease in serum albumin concentration [37,38]. Information regarding the dialyzer membrane was not collected in this study, although the only method of dialysis was HD for all patients included in this study. The differences in dialysis modalities and dialysis membranes may affect the hepatic oxygenation; hence, further studies are needed.”

Reference 37

Maduell F, Rodas L, Broseta JJ, Gomez M, Font MX, Molina A, et al. High-permeability alternatives to current dialyzers performing both high-flux hemodialysis and postdilution online hemodiafiltration. Artif Organs. 2019;43:1014-1021.

Reference 38

Cozzolino M, Magagnoli L, Ciceri P, Conte F, Galassi A. Effect of a medium cut-off (Theranova®) dialyser on haemodialysis patients: a prospective, cross-over study. Clin Kidney J. 2019: 14: 382-389.

---

## [Decision Letter · Decision Letter 1]

12 Oct 2021

Association between hepatic oxygenation on near-infrared spectroscopy and clinical factors in patients undergoing hemodialysis

PONE-D-21-15838R1

Dear Dr. Ookawara,

We’re pleased to inform you that your manuscript has been judged scientifically suitable for publication and will be formally accepted for publication once it meets all outstanding technical requirements.

Kind regards,

Miquel Vall-llosera Camps

Senior Editor

PLOS ONE

Reviewers' comments:

Reviewer's Responses to Questions

**Comments to the Author**

1. If the authors have adequately addressed your comments raised in a previous round of review and you feel that this manuscript is now acceptable for publication, you may indicate that here to bypass the “Comments to the Author” section, enter your conflict of interest statement in the “Confidential to Editor” section, and submit your "Accept" recommendation.

Reviewer #1: All comments have been addressed

Reviewer #2: All comments have been addressed

Reviewer #3: All comments have been addressed

2. Is the manuscript technically sound, and do the data support the conclusions?

Reviewer #1: Yes

Reviewer #2: (No Response)

Reviewer #3: Yes

3. Has the statistical analysis been performed appropriately and rigorously? 

Reviewer #1: Yes

Reviewer #2: (No Response)

Reviewer #3: Yes

4. Have the authors made all data underlying the findings in their manuscript fully available?

Reviewer #1: Yes

Reviewer #2: (No Response)

Reviewer #3: Yes

5. Is the manuscript presented in an intelligible fashion and written in standard English?

Reviewer #1: Yes

Reviewer #2: (No Response)

Reviewer #3: Yes

6. Review Comments to the Author

Reviewer #1: Thre authors adressed all comments satisfactorily and the changes made to the manuscript allow a better placement of it results in context.

Reviewer #2: (No Response)

Reviewer #3: This revised version has taken into accounts as possible reviewers' comments and explanations are now given on limitations of the study highlighted by reviewers analyses.

7. PLOS authors have the option to publish the peer review history of their article (what does this mean?). If published, this will include your full peer review and any attached files.

Reviewer #1: **Yes: **Thomas Reiter

Reviewer #2: No

Reviewer #3: No

---

## [Editor Report · Acceptance letter]

13 Oct 2021

PONE-D-21-15838R1 

Association between hepatic oxygenation on near-infrared spectroscopy and clinical factors in patients undergoing hemodialysis 

Dear Dr. Ookawara:

I'm pleased to inform you that your manuscript has been deemed suitable for publication in PLOS ONE. Congratulations! Your manuscript is now with our production department. 

Kind regards, 

on behalf of

Dr. Miquel Vall-llosera Camps 

Staff Editor

PLOS ONE